# A type I IFN-dependent DNA damage response regulates the genetic program and inflammasome activation in macrophages

Abigail J Morales[1], Javier A Carrero[2], Putzer J Hung[2], Anthony T Tubbs[2], Jared M Andrews[2], Brian T Edelson[2], Boris Calderon[2], Cynthia L Innes[3,4], Richard S Paules[3,4], Jacqueline E Payton[2], Barry P Sleckman[1]*

[1]Department of Pathology and Laboratory Medicine, Weill Cornell Medical College, New York City, United States; [2]Department of Pathology and Immunology, Washington University School of Medicine, St. Louis, United States; [3]Environmental Stress and Cancer Group, National Institute of Environmental Health Sciences, Durham, United States; [4]NIEHS Microarray Group, National Institute of Environmental Health Sciences, Durham, United States

**Abstract** Macrophages produce genotoxic agents, such as reactive oxygen and nitrogen species, that kill invading pathogens. Here we show that these agents activate the DNA damage response (DDR) kinases ATM and DNA-PKcs through the generation of double stranded breaks (DSBs) in murine macrophage genomic DNA. In contrast to other cell types, initiation of this DDR depends on signaling from the type I interferon receptor. Once activated, ATM and DNA-PKcs regulate a genetic program with diverse immune functions and promote inflammasome activation and the production of IL-1$\beta$ and IL-18. Indeed, following infection with *Listeria monocytogenes,* DNA-PKcs-deficient murine macrophages produce reduced levels of IL-18 and are unable to optimally stimulate IFN-$\gamma$ production by NK cells. Thus, genomic DNA DSBs act as signaling intermediates in murine macrophages, regulating innate immune responses through the initiation of a type I IFN-dependent DDR.

*For correspondence: bas2022@
med.cornell.edu

**Competing interests:** The authors declare that no competing interests exist.

## Introduction

Genomic DNA damage occurs upon exposure to genotoxic agents and during physiologic processes such as transcription and DNA replication. DNA double strand breaks (DSBs) are a dangerous form of DNA damage which, if incorrectly repaired, lead to chromosomal rearrangements and genome instability (*Jackson and Bartek, 2009*). Cells mount a canonical DNA damage response (DDR) to repair DSBs and protect the integrity of the genome (*Ciccia and Elledge, 2010*). In G1-phase cells, this DDR is initiated through the activation of ataxia telangiectasia mutated (ATM) and the catalytic subunit of the DNA-dependent protein kinase (DNA-PKcs), both members of the PI3-like family of serine threonine kinases (*Ciccia and Elledge, 2010*). These kinases phosphorylate downstream targets that drive the cellular response to DNA damage (*Ciccia and Elledge, 2010*). ATM is the primary kinase that orchestrates the DDR in G1-phase cells (*Ciccia and Elledge, 2010*). DNA-PKcs, by contrast, has fewer known DDR functions, but shares some downstream targets, and functions, with ATM (*Callén et al., 2009*).

The canonical DDR includes the activation of cell cycle checkpoints and, in G1-phase cells, the initiation of DSB repair by non-homologous end joining (NHEJ) (*Ciccia and Elledge, 2010*). However,

in some contexts, the DDR can regulate non-canonical cell type-specific responses. DNA DSBs generated by the RAG endonuclease during antigen receptor gene assembly activate a genetic program that is important for normal lymphocyte development (*Bednarski et al., 2012*, *2016*; *Bredemeyer et al., 2008*; *Helmink and Sleckman, 2012*). DNA damaging agents can also induce components of this genetic program in developing lymphocytes (*Bredemeyer et al., 2008*; *Innes et al., 2013*). DNA DSBs generated during immunoglobulin class switch recombination in mature B cells induce gene expression changes required for plasma cell differentiation (*Sherman et al., 2010*). RAG activity in lymphoid progenitors alters the function of mature T and NK cells, presumably through the generation of DSBs and activation of the DDR in these cells (*Karo et al., 2014*). Additionally, DDR activated by replication stress can promote the differentiation of polyploid macrophages that have developed in response to chronic inflammatory stimuli (*Herrtwich et al., 2016*).

Innate immune cells such as macrophages are among the first to respond to bacterial pathogens. They are activated by signals from Toll-like receptors (TLRs) that bind bacterial ligands and by interferon (IFN, type I or II) receptor signals (*Akira and Takeda, 2004*; *McNab et al., 2015*; *Schoenborn and Wilson, 2007*). Bacterial pathogens that enter the cytosol, such as *Listeria monocytogenes,* activate additional cytosolic pathways that lead to the production of type I IFN and the production of inflammatory cytokines IL-1$\beta$ and IL-18 through activation of the inflammasome (*Lamkanfi and Dixit, 2014*; *McNab et al., 2015*; *Witte et al., 2012*).

Once activated, macrophages produce genotoxic agents that kill bacterial pathogens, including reactive oxygen species (ROS) and reactive nitrogen species such as nitric oxide (NO) (*Mosser and Edwards, 2008*). ROS is generated rapidly through the activation of the NADPH oxidase. NO production, by contrast, is delayed due to a requirement to induce the expression of the *Nos2* gene, which encodes the nitric oxide synthase (*MacMicking et al., 1997*). *Nos2* expression is induced by TLR signals coupled with IFN receptor (type I or II) signals. Both ROS and NO have genotoxic properties and could conceivably damage host macrophage genomic DNA, initiating a DDR that could regulate macrophage functions in innate immune responses.

Here we establish functions for the DDR in regulating diverse innate immune responses in macrophages. We find that bone marrow-derived macrophages (BMDMs) activated by IFN-γ and LPS, the ligand for TLR4, or infected with *L. monocytogenes*, leads to the initiation of a DDR that requires type-I IFN signaling and regulates both the genetic program of activated macrophages and the production of IL-1$\beta$ and IL-18 by the inflammasome in these cells. These findings establish the DDR as an important signaling pathway in innate immune responses.

## Results

### Activated macrophages initiate an ATM- and DNA-PKcs-dependent DDR

The DDR was assayed in bone marrow-derived macrophages (BMDMs) by examining phosphorylation of the histone H2A variant, H2AX (forming γ-H2AX) or KAP-1, which are both substrates of ATM and DNA-PKcs. Treatment with IFN-γ and the TLR4 agonist, LPS, but neither agent alone, led to the initiation of a robust DDR in BMDMs (*Figure 1A*). Infection of BMDMs with the intracellular bacterial pathogen *L. monocytogenes* also led to a robust DDR, and IFN-γ augments, but is not required for, this response (*Figure 1B*). As with cultured BMDMs, LPS and IFN-γ treatment of primary macrophages isolated from the peritoneal cavity also activates a robust DDR (*Figure 1C*).

ATM is activated in BMDMs, as evidenced by the loss of KAP-1 phosphorylation in ATM-deficient (*Atm$^{-/-}$*) BMDMs treated with LPS and IFN-γ or infected with *L. monocytogenes* (*Figure 1D and E*). The formation of γ-H2AX in *Atm$^{-/-}$* BMDMs, however, indicates that additional DDR kinases must be activated (*Figure 1F and G*). WT BMDMs treated with the DNA-PKcs kinase inhibitor NU7026 and activated with LPS and IFN-γ exhibit robust KAP-1 and H2AX phosphorylation (*Figure 1D*). Inhibition of DNA-PKcs kinase activity in *Atm$^{-/-}$*BMDMs, however, abrogated H2AX phosphorylation in response to LPS and IFN-γ, demonstrating that both ATM and DNA-PKcs are activated in response to LPS and IFN-γ (*Figure 1F*).

*Scid* mice are deficient in DNA-PKcs due to a point mutation in the DNA-PKcs gene that results in low-level expression of a non-functional DNA-PKcs protein (*Blunt et al., 1996*). *Scid* BMDMs

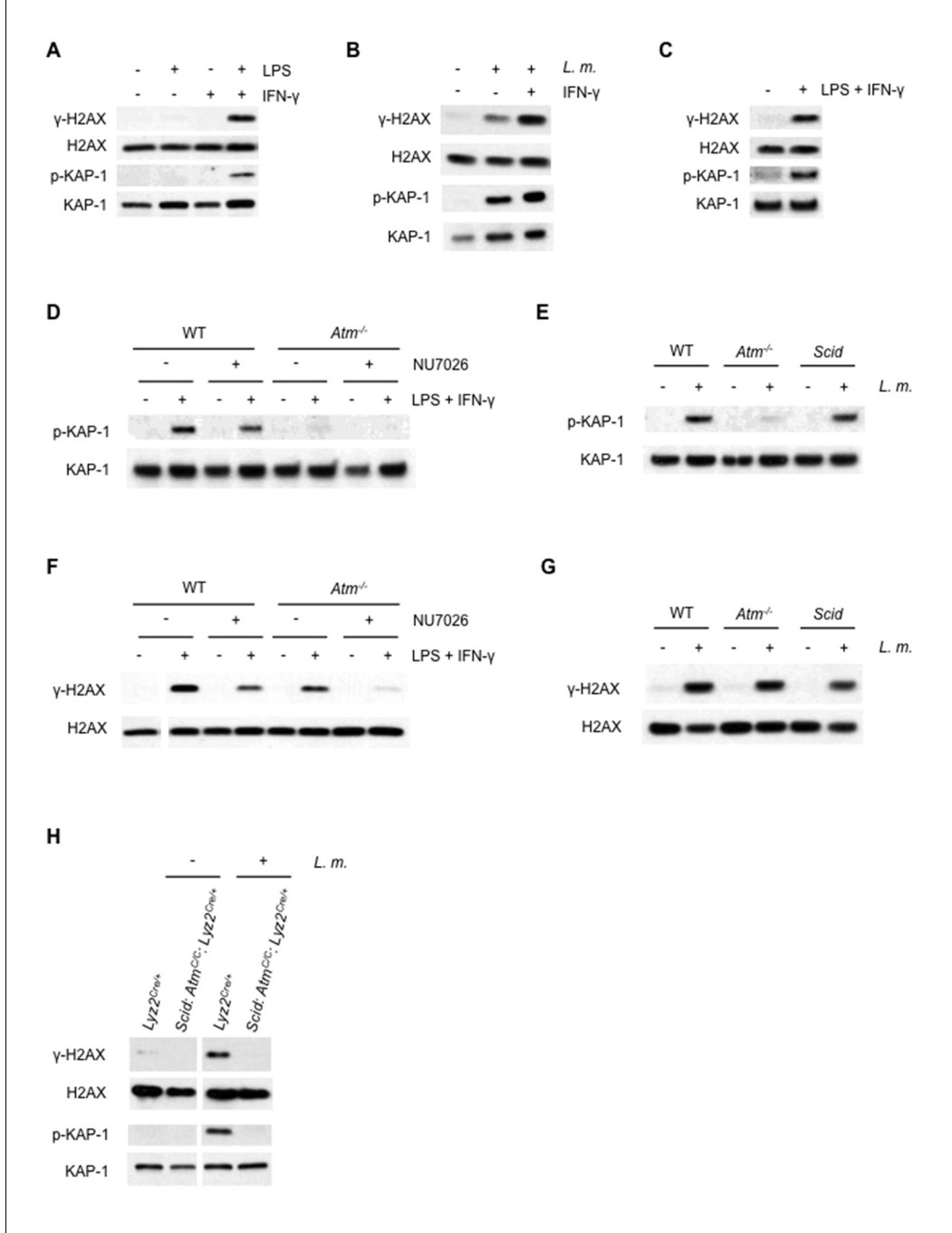

**Figure 1.** ATM and DNA-PKcs-dependent DDR activation in macrophages. (**A–C**) Western blotting for γ-H2AX, H2AX, phosphorylated KAP-1 (p-KAP-1) and KAP-1 in whole cell lysates from (**A**) wild type (WT) BMDMs after no stimulation (-) or stimulation (+) with LPS, IFN-γ, or both for 9 hr. (**B**) WT BMDMs after no infection (-) or infection (+) with *L. monocytogenes* (*L. m.*) either in the presence (+) or absence (-) of IFN-γ for 9 hr. (**C**) WT peritoneal macrophages after no stimulation (-) or stimulation (+) with LPS + IFN-γ for 24 hr. (**D, F**) Western blotting of p-KAP-1 and KAP-1 (**D**) and γ-H2AX and H2AX (**F**) in whole cell lysates from WT and *Atm⁻/⁻* BMDMs after no treatment (-) or treatment (+) with LPS + IFN-γ for 9 hr. in the presence (+) or absence (-) of NU7026. (**E, G**) Western blotting of p-KAP-1 and KAP-1 (**E**) and γ-H2AX and H2AX (**G**) in whole cell lysates from WT, *Atm⁻/⁻*, and *Scid* BMDMs after no infection (-) or infection (+) with *L. m.* for 24 hr. (**H**) Western blotting for γ-H2AX, H2AX, p-KAP-1, and KAP-1 in whole cell lysates from *Lyz2^Cre/+* and *Scid:Atm^C/C:Lyz2^Cre/+* BMDMs after no infection (-) or infection (+) with *L. m.* for 24 hr. Data are representative of 2–5 independent experiments. Blank spaces in (**F**) and (**H**) indicate that blots have been cropped.

*Figure 1 continued on next page*

*Figure 1 continued*

The following figure supplements are available for figure 1:

**Figure supplement 1.** ATM is efficiently deleted in BMDM.

**Figure supplement 2.** *L. monocytogenes* infection induces G1 arrest.

infected with *L. monocytogenes* exhibit robust KAP-1 and H2AX phosphorylation (*Figure 1G and E*). To generate BMDMs deficient in both ATM and DNA-PKs, *Scid* mice homozygous for a conditionally targeted ATM allele (*Atm^C*) and heterozygous for a Cre knock-in at the lysozyme M (*Lyz2*) locus (*Scid:Atm^{C/C}:Lyz2^{Cre/+}*) were generated (*Clausen et al., 1999*). Mice deficient in ATM and DNA-PKcs exhibit early embryonic lethality (*Sekiguchi et al., 2001*). *Scid:Atm^{C/C}:Lyz2^{Cre/+}* mice are viable, however, and BMDMs from these mice are deficient in DNA-PKcs (*Scid*) and have deleted both ATM alleles (*Figure 1—figure supplement 1*). Like WT BMDMs, those that express Cre (*Lyz2^{Cre/+}* BMDMs) exhibit robust γ-H2AX formation and KAP-1 phosphorylation in response to infection with *L. monocytogenes* (*Figure 1H*). Whereas BMDMs with isolated deficiencies of ATM or DNA-PKcs exhibit a robust DDR, those with deficiencies in both ATM and DNA-PKcs (*Scid:Atm^{C/C}:Lyz2^{Cre/+}*) exhibit a near-complete abrogation of γ-H2AX and p-KAP-1 formation in response to infection with *L. monocytogenes* (*Figure 1H*). Though uninfected *Lyz2^{Cre/+}* and *Scid:Atm^{C/C}:Lyz2^{Cre/+}* BMDM are cycling, both undergo G1 arrest after infection with *L. monocytogenes* (*Figure 1—figure supplement 2*). We conclude that both ATM and DNA-PKcs are activated in BMDMs by LPS and IFN-γ or infection with *L. monocytogenes,* and that these kinases can have unique (ATM phosphorylation of KAP-1) or overlapping (ATM or DNA-PKcs phosphorylation of H2AX) functions.

## The DDR is activated by genomic DNA DSBs

ATM can be directly activated by oxidizing agents, however, several lines of evidence demonstrate that the DDR in activated macrophages is initiated primarily by genomic DNA DSBs (*Guo et al., 2010*). The neutral comet assay, which quantifies genomic DSBs in single cells, revealed a significant increase in the Olive Tail Moment (measure of DNA DSBs) in BMDM after infection with *L. monocytogenes* or treatment with LPS and IFN-γ (*Figure 2A*). γ-H2AX forms in chromatin flanking genomic DNA DSBs, leading to discrete nuclear foci that can be detected by immunostaining (*Rogakou et al., 1999*). BMDMs infected with *L. monocytogenes* exhibit an increase in γ-H2AX nuclear foci relative to uninfected cells, indicative of genomic DNA DSB formation (*Figure 2B*, *Figure 2—figure supplement 1*). That these DSBs activate the DDR is evidenced by the analyses of BMDMs deficient in either DNA DSB repair or the Mre11, Rad50, Nbs1 (MRN) complex, which is required to sense DNA DSBs and activate ATM (*Ciccia and Elledge, 2010*). Deficiency in DNA Ligase IV, which is required for DSB repair by NHEJ, leads to embryonic lethality (*Frank et al., 1998*). However, mice homozygous for a conditionally targeted DNA Ligase IV allele and heterozygous for *Lyz2^{Cre}* (*Lig4^{loxP/loxP}:Lyz2^{Cre/+}*) are viable and have BMDMs deficient in DNA Ligase IV (data not shown). As compared to *Lyz2^{Cre/+}* BMDMs (normal NHEJ), infection of *Lig4^{loxP/loxP}:Lyz2^{Cre/+}* BMDMs (loss of NHEJ) with *L. monocytogenes* leads to augmented γ-H2AX formation (*Figure 2C*). Mice homozygous for a hypomorphic Mre11 allele (*Mre11^{ATLD1/ATLD1}*) exhibit diminished ATM activation in response to DNA DSBs, and BMDMs derived from these mice exhibit a significant reduction in the ATM-dependent phosphorylation of KAP-1 in response to LPS and IFN-γ (*Figure 2D*) (*Theunissen et al., 2003*). Taken together, these findings demonstrate that the initiation of a DDR in activated macrophages occurs primarily through the generation of genomic DNA DSBs.

## Nitric oxide initiates DDR

Macrophage activation leads to production of ROS and NO, both of which have genotoxic properties that could generate DNA DSBs. We treated BMDMs with a cell-permeable superoxide scavenger, MnTMPyP, or an inhibitor of the inducible nitric oxide synthase, aminoguanidine hemisulfate (AGHS). Treatment with AGHS, but not MnTMPyP, prevented γ-H2AX formation in response to LPS and IFN-γ, suggesting that NO, and not ROS, initiates DDR in activated BMDMs (*Figure 2E*). In

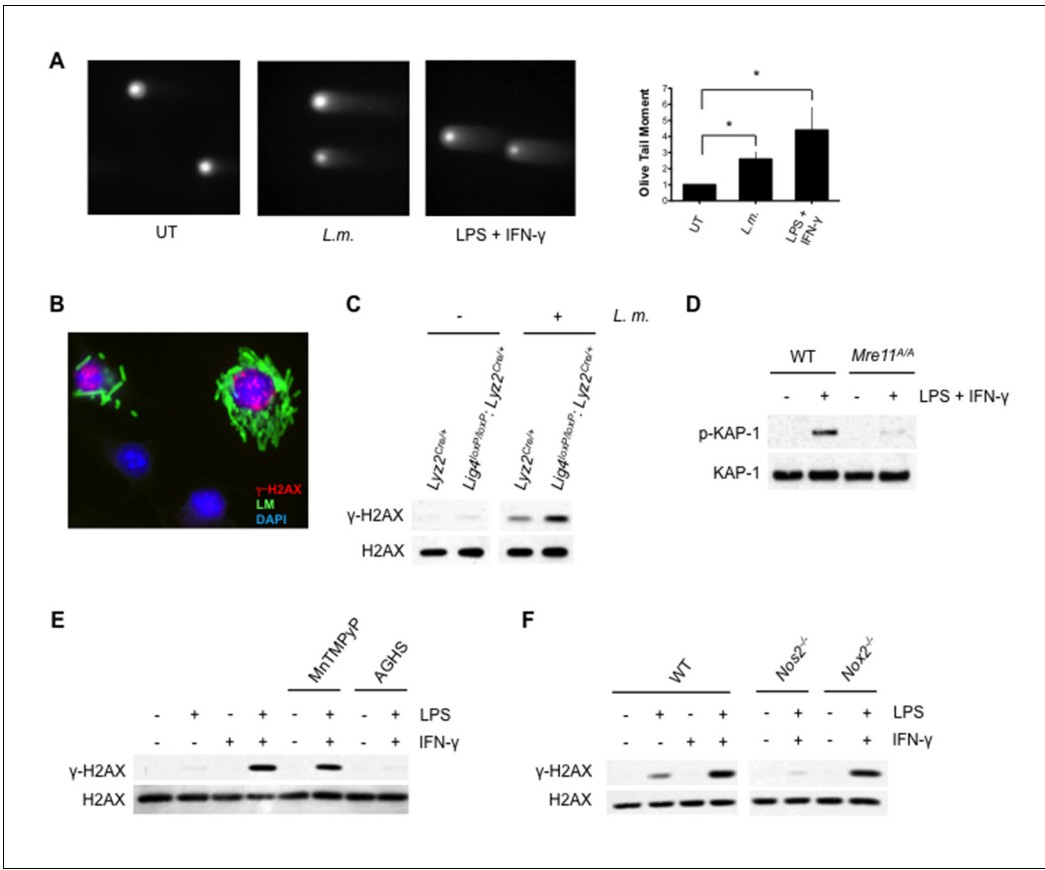

**Figure 2.** Activation of DDR by DNA DSBs. (**A**) Neutral Comet assay and Olive Tail Moment quantification of BMDMs left untreated (UT) or after infection with *L. m.* or LPS + IFN-γ for 12 hr. (**B**) Immunofluorescence for *L. m.* (green) and γ-H2AX (red) in WT BMDMs 9 hr. post-infection. Nuclei are revealed by DAPI (blue). (**C**) Western blotting of DNA Ligase IV-deficient (*Lig4*<sup>loxP/loxP</sup>: *Lyz2*<sup>Cre/+</sup>) or *Lyz2*<sup>Cre/+</sup> BMDMs after no infection (-) or infection (+) with *L. m.* for 24 hr. (**D**) Western blotting of WT or *Mre11*<sup>ATLD1/ATLD1</sup> BMDMs after no stimulation (-) or stimulation (+) with LPS + IFN-γ for 9 hr. (**E, F**) Western blot analysis of (**E**) WT BMDMs after no stimulation (-) or stimulation (+) with LPS + IFN-γ for 9 hr. in the presence or absence of MnTMPyP or AGHS (**F**) WT, iNOS-deficient (*Nos2*<sup>-/-</sup>), or NADPH oxidase-deficient (*Nox2*<sup>-/-</sup>) BMDMs after no stimulation (-) or stimulation (+) with LPS + IFN-γ for 9 hr. Data are representative of two or more independent experiments. Quantitation in (**A**) is the mean and SEM of three independent experiments where ≥ 50 tails were analyzed for each condition and Olive Tail Moment of treated BMDMs is expressed as a ratio to untreated BMDMs. *p<0.05 (Student's paired t-test). Blank spaces in (**C**) and (**FE**) indicate that blots have been cropped.

The following figure supplement is available for figure 2:

**Figure supplement 1.** *L.monocytogenes*-infected BMDM exhibit discrete γ-H2AX foci.

agreement with this finding, BMDMs deficient in the nitric oxide synthase (*Nos2*<sup>-/-</sup>) exhibit diminished γ-H2AX in response to LPS and IFN-γ, whereas loss of gp91<sup>phox</sup> (*Nox2*<sup>-/-</sup>), an essential catalytic subunit of the NADPH oxidase, has no effect on γ-H2AX formation (*Figure 2F*) (*Laubach et al., 1995*; *Pollock et al., 1995*). Thus, activation of the DDR in macrophages depends on NO production.

## Type I IFN receptor signals drive the DDR in BMDMs

The requirement for both LPS and IFN-γ indicates that both TLR and IFN receptor signaling may be needed to activate the DDR in macrophages (*Figure 1A*). Indeed, after infection of BMDMs with *L. monocytogenes*, DDR activation depends on TLR signaling, as BMDMs deficient in MyD88, a critical TLR adaptor protein, exhibit diminished γ-H2AX formation (*Figure 3A*). However, infection of wild

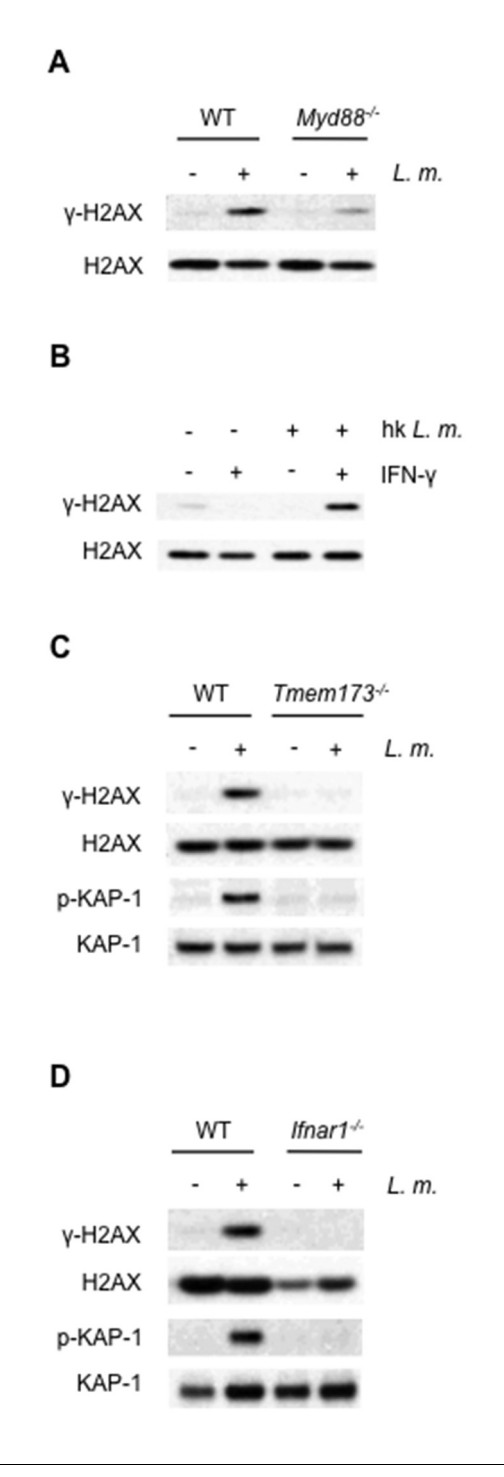

**Figure 3.** The DDR depends on TLR and interferon signaling cascades. (A,B) Western blotting for γ-H2AX and H2AX in whole cell lysates from (A) WT or *Myd88*[-/-] BMDMs after no infection (-) or infection (+) with *L. monoctytogenes* (*L. m.*) for 24 hr. (B) WT BMDMs after no treatment (-) or treatment (+) with IFN-γ, heat-killed *L. monocytogenes*, or both for 9 hr. (C, D) Western blotting for γ-H2AX, H2AX, p-KAP-1, and KAP-1 in whole cell lysates from WT, *Tmem173*[-/-] (C) or
*Figure 3 continued on next page*

type BMDMs with *L. monocytogenes* leads to robust DDR activation without the addition of IFN-γ (*Figure 1B*). In this regard, type I IFN produced upon entry of *L. monocytogenes* into the cytosol may provide IFN signals required for DDR initiation (*Witte et al., 2012*). In agreement with this notion, heat-killed *L. monocytogenes*, which activates TLR signaling but does not enter the cytosol and stimulate type I IFN production, does not initiate a DDR in the absence of exogenous IFN (*Figure 3B*). Moreover, BMDMs deficient in STING (*Tmem173*[-/-]), a cytosolic protein required to stimulate type I IFN production downstream of *L. monocytogenes* DNA, do not initiate a DDR after infection with *L. monocytogenes* (*Figure 3C*). Finally, DDR signaling is not observed in type I IFN receptor-deficient (*Ifnar1*[-/-]) BMDMs infected with *L. monocytogenes* (*Figure 3D*).

TLR and IFN receptor signaling may be required primarily to induce the DNA damaging agent, NO. Though *Ifnar1*[-/-] BMDMs infected with *L. monocytogenes* do not produce NO, addition of IFN-γ rescues NO production to wild type levels (*Figure 4A*). This, however, does not lead to a robust DDR (*Figure 4B*). Thus, type I IFN signaling is required to promote the DDR in BMDMs beyond its role in stimulating NO production. The DDR is also attenuated in *Ifnar1*[-/-] BMDMs treated with LPS and IFN-γ (data not shown). Type I IFN does not augment DNA DSB generation, as evidenced by the equivalent levels of DNA DSBs observed by neutral comet assay analysis of BMDMs treated with the DNA DSB-inducing agent bleomycin in the presence or absence of type I IFN (IFN-*β*) (*Figure 4C*). However, a robust DDR to bleomycin is only observed in BMDMs treated with IFN-*β* (*Figure 4D*). In contrast to macrophages, mouse embryonic fibroblasts (MEFs) treated with bleomycin exhibit a robust DDR that is not augmented by the addition of either IFN-*β* or IFN-γ (*Figure 4D and E*). Thus, unlike other cell types, the DDR in macrophages depends on type I IFN signaling. This is not due to an effect of type I IFN signaling on cell cycle parameters or the expression of key DDR proteins (Mre11, Rad50, Nbs1, Ku70, Ku80, ATR, DNA-PKcs, ATM and H2AX) in BMDMs (*Figure 4—figure supplements 1* and *2*). Notably, many of these proteins exhibit substantially lower expression in BMDMs as compared to MEFs (see Discussion).

*Ifnar1⁻ᐟ⁻* (**D**) BMDMs after no infection (-) or infection with *L. m.* for 24 hr. Data are representative of 2–4 independent experiments.

## DDR regulates the genetic program of macrophages

We performed gene expression profiling on *Lyz2*$^{Cre/+}$ and *Scid:Atm*$^{C/C}$:*Lyz2*$^{Cre/+}$ BMDMs before and after infection with *L. monocytogenes* (*Figure 5A* and *Figure 5—source data 1*). The expression of 322 genes was induced (≥4 fold) after infection of *Lyz2*$^{Cre/+}$ BMDMs with *L. monocytogenes* (*Figure 5A* and

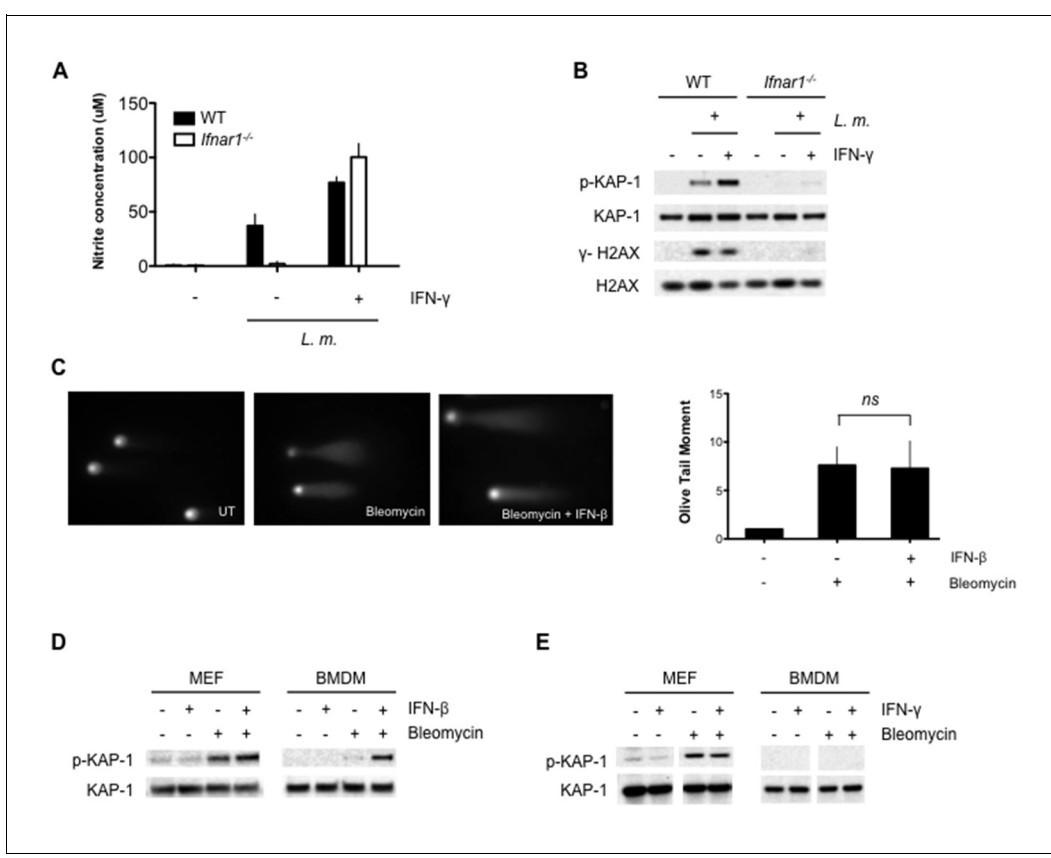

**Figure 4.** Optimal DDR depends on type I interferon signaling. (**A**) Nitrite concentration in culture supernatants collected from WT and *Ifnar1-/-* BMDMs after no infection (-) or infection (+) with *L. m.* for 24 hr. in the presence or absence of IFN-γ. Data are a compilation of three independent experiments and depict mean and SEM. (**B**) Western blotting for γ-H2AX, H2AX, p-KAP-1, and KAP-1 in whole cell lysates from WT and *Ifnar1-/-* BMDMs after no infection (-) or infection (+) with *L. m.* for 24 hr. in the presence or absence of IFN-γ . (**C**) Neutral Comet assay and Olive Tail Moment quantification of BMDMs left untreated (UT) or after treatment with bleomycin in the presence or absence of type I interferon (IFN-*β*). (**D, E**) Western blotting for p-KAP-1 and KAP-1 of WT mouse embryonic fibroblasts (MEFs) or BMDMs after no stimulation (-) or stimulation with bleomycin (1 ug/mL), (**D**) IFN-*β* (100 U/mL), or bleomycin + IFN-*β* for 6 hr. (**E**) IFN-γ (10 ng/mL), or bleomycin + IFN-γ for 6 hr. Data are representative of 2-4 independent experiments. Quantitation in (**C**) is the mean and SEM of three independent experiments where ≥50 tails were analyzed for each condition and Olive Tail Moment of treated is expressed as a ratio to untreated. *ns* = not significant. Blank space in (**E**) indicates that blot has been cropped

The following figure supplements are available for figure 4:

**Figure supplement 1.** Type I IFN does not affect the cell cycle in BMDMs.

**Figure supplement 2.** Type I IFN does not impact expression of key DDR factors.

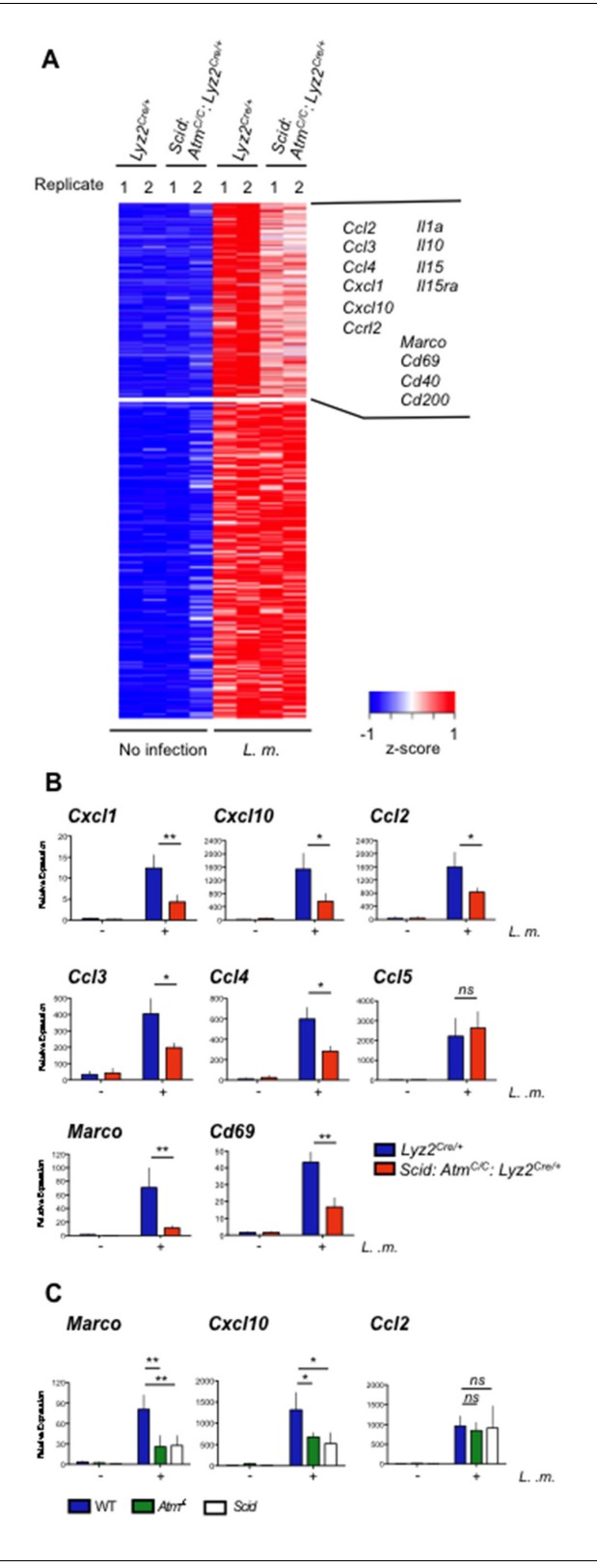

**Figure 5.** ATM and DNA-PKcs regulate the genetic program of activated macrophages. (**A**) Heat map showing genes that are significantly up-regulated (≥4 fold, adj. p-value<0.001) in *L. m.*-infected *Lyz2*$^{Cre/+}$and *Scid:Atm* $^{C/C}$: *Lyz2*$^{Cre/+}$ BMDMs relative to no infection. Red indicates relatively higher expression; blue indicates relatively lower expression. Genes with increased expression (>1.25 fold) in *Lyz2*$^{Cre/+}$controls relative to *L. m.*-infected *Scid:Atm* $^{C/}$

*Figure 5 continued on next page*

*Figure 5 continued*

$^{C}$:Lyz2$^{Cre/+}$ BMDMs are clustered in the upper right hand corner of the heat map. Select genes in this group are indicated. Shown are two biological replicates (1 and 2) for each condition and genotype. (B,C) Quantitative real-time PCR (RT-PCR) of gene expression in uninfected (-) and *L. m.*-infected (+) (B) *Scid:Atm* $^{C/C}$:Lyz2$^{Cre/+}$ and *Lyz2*$^{Cre/+}$ BMDMs and (C) *Atm*$^{-/-}$, *Scid*, and WT BMDMs. Data are the mean and SEM of three or more independent experiments. *p<0.05, **p<0.01, *ns* = not significant.

The following source data is available for figure 5:

**Source data 1.** Gene expression changes induced by *L. monocytogenes* infection.

*Figure 5—source data 1*). Of these, 128 exhibited greater (≥1.25 fold) induction in *Lyz2*$^{Cre/+}$ BMDMs infected with *L. monocytogenes* relative to infected *Scid:Atm*$^{C/C}$:Lyz2$^{Cre/+}$ BMDMs, indicating that their expression is regulated by ATM and/or DNA-PKcs (*Figure 5A* and *Figure 5—source data 1*). These genes encode proteins with diverse functions in the immune response, including several cytokines, chemokines, and cell surface proteins such as the class A scavenger receptor MARCO and CD69, which has a role in cell localization and migration (*Figure 5A and B*) (*Kraal et al., 2000*; *Schwab and Cyster, 2007*). The DDR in *L. monocytogenes*-infected BMDMs regulates the expression of many (*Cxcl1, Cxcl10, Ccl2, Ccl3* and *Ccl4*), but not all (*Ccl5*), chemokine genes that are induced in activated macrophages (*Figure 5A and B*). The analysis of BMDMs deficient in ATM (*Atm*$^{-/-}$) or DNA-PKcs (*Scid*) revealed that both of these kinases can function to regulate gene expression (*Figure 5C, Marco* and *Cxcl10*), whereas in other cases the activity of either of these kinases is adequate (*Figure 5C, Ccl2*). We conclude that once activated by DNA DSBs, ATM and DNA-PKcs regulate the functional genetic program of activated macrophages.

## Regulation of inflammasome activation by ATM and DNA-PKcs

Entry of *L. monocytogenes* into the cytosol activates the NLRP3 and AIM2 inflammasomes, leading to the activation of the caspase 1 protease and cleavage of pro-IL-1$\beta$ and pro-IL-18 to form active IL-1$\beta$ and IL-18, respectively (*Lamkanfi and Dixit, 2014*; *von Moltke et al., 2013*; *Witte et al., 2012*). Loss of both ATM and DNA-PKcs (*Scid:Atm*$^{C/C}$:Lyz2$^{Cre/+}$) leads to a significant reduction in IL-1$\beta$ production in response to infection of BMDMs with *L. monocytogenes* (*Figure 6A*). In contrast, BMDMs with isolated deficiencies in ATM or DNA-PKcs exhibit only a mild reduction in IL-1$\beta$ production (data not shown). ROS levels in resting ATM-deficient macrophages have been implicated in blunting inflammasome responses (*Erttmann et al., 2016*). However, we do not observe any notable differences in ROS levels among the different macrophages examined here (*Figure 6—figure supplement 1*). ATM and DNA-PKcs are not required to promote pro-IL-1$\beta$ gene expression, nor are they required for optimal expression of the *Asc, Nlrp3*, or *Aim2* genes (*Figure 6—figure supplement 2*). Rather, after infection with *L. monocytogenes, Scid:Atm*$^{C/C}$:Lyz2$^{Cre/+}$ BMDMs are unable to efficiently convert inactive pro-caspase 1 to active caspase 1 (p20) (*Figure 6B*). This requirement for ATM and DNA-PKcs is not specific to *L. monocytogenes,* as AIM2 inflammasome activation by LPS and poly dA:dT also depends on an intact DDR (*Figure 6C*). Moreover, activation of the NLRP3 inflammasome with LPS and either nigericin or monosodium urate (MSU) crystals is also defective in *Scid:Atm*$^{C/C}$:Lyz2$^{Cre/+}$ BMDMs as evidenced by reduced IL-1$\beta$ production in response to these stimuli (*Figure 6D*).

In contrast to IL-1$\beta$, production of IL-18 relies primarily on DNA-PKcs, as *Scid* BMDMs produce significantly less IL-18 after *L. monocytogenes* infection (*Figure 7A*). DNA-PKcs is not required to promote pro-IL-18 gene expression (*Figure 7—figure supplement 1*). IL-12 and IL-18 stimulate T cells and NK cells to produce IFN-γ (*Akira, 2000*; *Hsieh et al., 1993*; *Okamura et al., 1995*). As compared to wild type BMDMs, *Scid* BMDMs infected with *L. monocytogenes* were not effective at simulating IFN-γ production by co-cultured NK cells (*Figure 7B*). *Scid* and wild type BMDMs both induce IL-12 (p40) gene expression upon infection with *L. monocytogenes* and the addition of exogenous IL-18 rescues IFN-γ production by NK cells co-cultured with *L. monocytogenes*-infected *Scid* BMDMs (*Figure 7B* and *Figure 7—figure supplement 1*). Thus, the inability of *Scid* BMDMs to induce NK cells to make IFN-γ is due to the role of DNA-PKcs stimulating IL-18 production. We concluded that ATM and DNA-PKcs regulate inflammasome function in response to a broad variety of

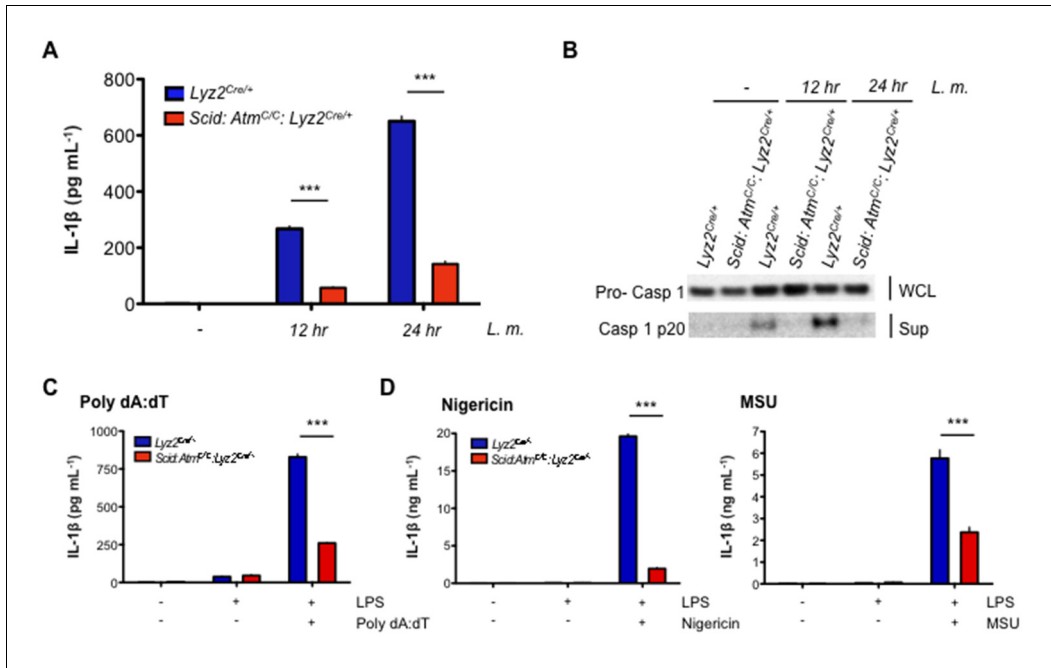

**Figure 6.** DDR regulates inflammasome activation. (**A**) IL-1$\beta$ protein concentration as determined by ELISA in supernatants from uninfected (-) and *L. m.*-infected (+) *Scid:Atm$^{C/C}$:Lyz2$^{Cre/+}$* and *Lyz2$^{Cre/+}$* BMDMs 12 and 24 hr. post-infection. Data are mean and SEM of technical replicates and are representative of four independent experiments. ***p<0.0001. (**B**) Western blot analysis of pro-caspase 1 in whole cell lysates (WCL) and active caspase 1 (p20) in supernatants (Sup) from *Lyz2$^{Cre/+}$* and *Scid:Atm$^{C/C}$:Lyz2$^{Cre/+}$* BMDMs 12 and 24 hr. post-*L. m.* infection. Data are representative of two independent experiments. (**C,D**) IL-1$\beta$ protein concentration by ELISA in supernatants from *Lyz2$^{Cre/+}$* and *Scid:Atm$^{C/C}$:Lyz2$^{Cre/+}$* BMDMs left untreated (-) or after treatment with LPS (200 ng/mL) for 12 hr and in the presence or absence of (**C**) poly(dA:dT) (5 ug/mL) for 12 hr. (**D**) nigericin (5 uM) for 30 min. or monosodium urate crystals (MSU, 50 ug/mL) for 12 hr. Data are mean and SEM of technical replicates and are representative of two or three independent experiments. ***p<0.0001.

The following figure supplements are available for figure 6:

**Figure supplement 1.** Mitochondrial ROS levels are equivalent in DDR-deficient and –sufficient BMDM.

**Figure supplement 2.** Gene expression of inflammasome components.

activators. Moreover, these two kinases are differentially required for the production of IL-1$\beta$ and IL-18 by the inflammasome.

## Discussion

Here we establish that DDR signaling pathways regulate inflammasome activation and transcriptional programs with innate immune functions in macrophages. In activated macrophages, the DDR is initiated by genomic DNA DSBs that can be generated by NO produced in response to TLR and type I IFN signals (*Figure 7C*). However, in addition to stimulating NO production, we find that type I IFN signals are required for optimal DDR activation in macrophages. This is not due to a requirement for type I IFN signaling in the generation of DSBs. Rather, we find that type I IFN signals are required to augment the DDR in macrophages, but not MEFs, once DSBs are generated. Our findings establish a mechanistic link between DNA DSB generation, DDR signaling and innate immune responses mediated by macrophages.

We have primarily analyzed macrophages activated either with LPS and IFN-$\gamma$ or after infection with *L. monocytogenes.* Most bacteria initiate TLR signals and stimulate type I IFN production, leading to NO production, which should generate DNA DSBs and activate a DDR in macrophages. Thus,

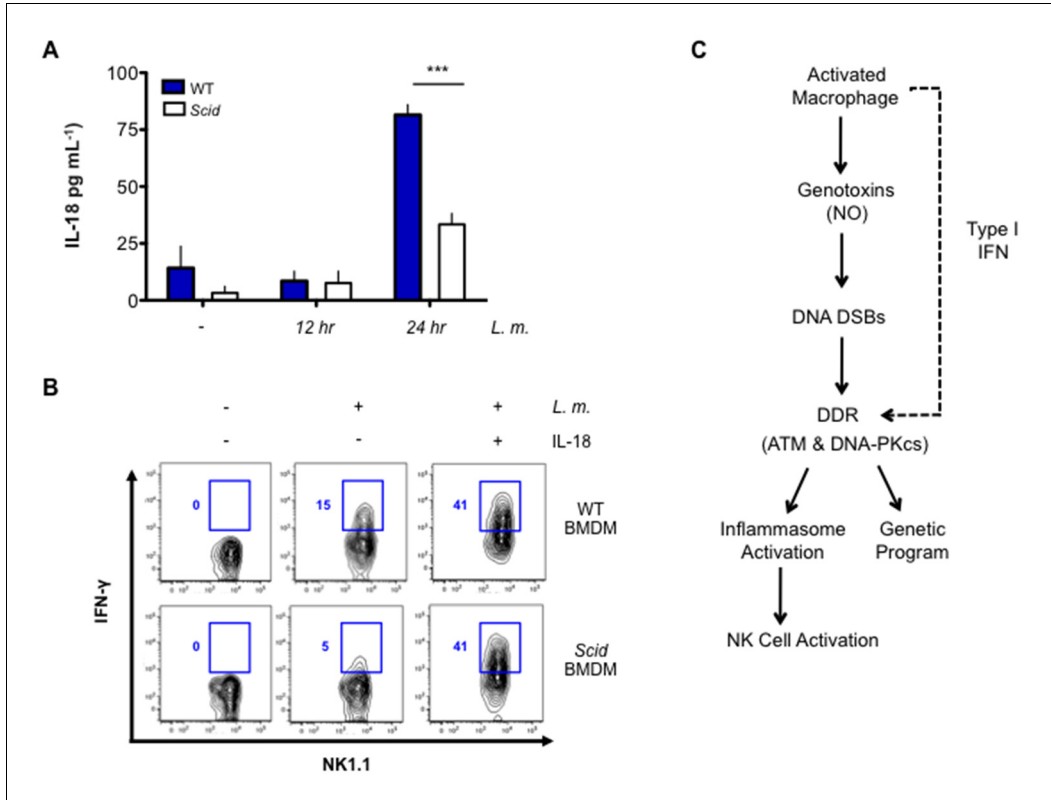

**Figure 7.** DNA-PKcs regulates IL-18 production and NK cell activation. (**A**) IL-18 protein concentration as determined by ELISA in supernatants from uninfected (-) and *L. m.*-infected WT and *Scid* BMDMs 12 and 24 hr. post-infection. Data are mean and SEM of technical replicates and are representative of three independent experiments. ***$p \leq 0.0002$. (**B**) Flow cytometric analysis of intracellular IFN-γ production by WT splenic NK cells co-cultured with uninfected or *L. m.*-infected WT or *Scid* BMDM with (+) or without (-) the addition of IL-18 for 15 hr. Data are representative of four independent experiments. (**C**) Schematic showing the regulation of macrophage functions by DDR.
The following figure supplement is available for figure 7:

**Figure supplement 1.** Il18 and Il12 gene expression is equivalent in WT and *Scid* BMDM after *L.m.* infection.

we reason that the DDR will be activated, and regulate macrophage functions, in response to a broad variety of bacterial pathogens and other activating stimuli. Indeed, we find that the DDR augments inflammasome function in macrophages in response to LPS and nigericin or LPS and MSU, which activate the NLRP3 inflammasome, and to LPS and poly dA:dT, which activates the AIM2 inflammasome.

Genotoxic agents other than NO could also cause DNA damage and initiate a DDR. In this regard, ROS initiates a DDR in a human cervical cell line infected with the intracellular pathogen *Chlamydia trachomatis* (*Chumduri et al., 2013*). However, we did not find that ROS initiated a DDR in macrophages under the conditions examined here. Perhaps this is due to the robust activity of enzymes, such as superoxide dismutase or catalase, that inactivate ROS in macrophages before it can cause DNA damage (*Nathan and Cunningham-Bussel, 2013*). The activation of DDR in macrophages by NO, and not ROS, has important temporal implications for the effect of the DDR on macrophage function. Whereas ROS is produced immediately upon macrophage activation, NO production is delayed due to the requirement for induction of *Nos2* gene expression and production of nitric oxide synthase (*MacMicking et al., 1997*; *Nunes et al., 2013*). Thus, NO-induced DNA damage, the initiation of a DDR, and its physiologic effects on macrophage function will be delayed relative to macrophage activation. Though the mechanism by which NO induces DNA damage

remains unclear, NO can react with other species, giving rise to agents such as $N_2O_3$, which can cause DNA crosslinking and strand breaks, and the highly reactive peroxynitrite, which rapidly reacts with DNA and can cause DNA breaks and mutations (*Anand and Stamler, 2012*; *Radi, 2013*; *Tamir et al., 1996*). Though these lesions likely occur at random in macrophage DNA, it is certainly possible that there are regions within the genome that are more susceptible to genotoxic damage by nitric oxide and its metabolic derivatives.

It is possible that DNA DSBs could be generated, and a DDR initiated, in activated macrophages in the absence of NO. In this regard, pathogens can produce agents that can directly or indirectly cause DNA damage (*Guerra et al., 2011*; *Weitzman and Weitzman, 2014*). Cytolethal Distending Toxins (CDT), produced by some bacteria, have phosphodiesterase activity and can cause DNA DSBs (*Alaoui-El-Azher et al., 2010*; *Fedor et al., 2013*; *Frisan et al., 2003*; *Guerra et al., 2011*; *Jinadasa et al., 2011*; *Nesić et al., 2004*; *Weitzman and Weitzman, 2014*). Some strains of bacteria produce the genotoxic agent colibactin (*Cuevas-Ramos et al., 2010*; *Nougayrède et al., 2006*). Moreover, some bacteria produce agents that can directly modulate the DDR and its activities in response to these pathogens (*Weitzman and Weitzman, 2014*). Thus, different bacteria may have evolved ways to induce DNA DSBs and to modulate the DDR.

The sensing of bacterial and viral DNA in the cytosol of infected cells is a key initiating event in inflammasome activation and in the production of type I IFN, which is required for a robust DDR. The DDR is initiated by proteins that directly bind to DNA DSBs such as Ku70 and Ku80, which activate DNA-PKcs and the MRN complex, which activates ATM (*Ciccia and Elledge, 2010*). These proteins have been shown to bind to DNA DSBs in a sequence-independent fashion and are capable of binding mammalian, bacterial and viral DNA. In some settings, several of these proteins have been implicated as sensors for type I IFN production in response to cytosolic bacterial or viral DNA (*Chatzinikolaou et al., 2014*; *Lilley et al., 2007*) or can couple with known innate immune adaptors to stimulate the production of pro-IL-1$\beta$ downstream of NF-$\kappa$B activation (*Roth et al., 2014*). We show here, however, that these proteins can regulate macrophage function by activating a DDR downstream of genomic DNA DSBs. In activated macrophages, DDR activation by LPS and IFN-$\gamma$ depends on the production of type I IFN and on the MRN component, Mre11. In this setting, Mre11 functions to activate DDR to genomic DSBs and not as a sensor of bacterial or viral DNA. Moreover, we show that the activation of the inflammasome by MSU and nigericin depends on ATM and DNA-PKcs, despite the fact that neither of these stimuli include pathogen DNA.

In most cell types, ATM is the predominant kinase that mediates DDR. In macrophages, however, we find that full activation of the DDR depends on both ATM and DNA-PKcs. Indeed, ATM and DNA-PKcs have unique functions required for the expression of some genes and overlapping functions that regulate others. The DDR regulates the expression of a variety of chemokine genes in activated macrophages. This includes CCL2, which mediates inflammatory monocyte trafficking into peripheral tissues, CCL3 (MIP-1$\alpha$), CCL4 (MIP-1$\beta$), and CXCL1, which regulate the migration of innate immune cells such as macrophages, neutrophils, and NK cells (*Griffith et al., 2014*; *Kuziel et al., 1997*). NF-$\kappa$B is a critical transcription factor that regulates chemokine gene expression downstream of pattern recognition receptors as well as IL-1 and TNF receptors. DNA damage can also lead to the ATM-dependent activation of NF-$\kappa$B (*Huang et al., 2003*; *Wu et al., 2006*). Thus, it is possible that DDR in activated macrophages regulate chemokine gene expression by augmenting NF-$\kappa$B activation. Moreover, *Cxcl1* expression is regulated by NF-$\kappa$B and poly(ADP-ribose) polymerase (PARP)$-$1, which is activated by DNA damage and has a broad range of functions in DDR (*Amiri and Richmond, 2003*; *Nirodi et al., 2001*; *Rouleau et al., 2010*). Thus, in activated macrophages, the initiation of a DDR may regulate the genetic program through multiple signaling and transcription pathways.

The DDR-regulated genetic program in activated macrophages shares functional similarities with the ATM-dependent genetic program in developing pre-B lymphocytes (*Bednarski et al., 2016*; *Bredemeyer et al., 2008*). CD69, which has an established role in cell homing and migration, is regulated by DDR in both pre-B cells undergoing antigen receptor rearrangement and in activated macrophages. Indeed, many of the DDR-regulated genes in both developing lymphocytes and activated macrophages have an established role in cell migration. Optimal expression of *Ccl2*, *Ccl3*, *Ccl4*, and *Cxcl1* depends on DDR signaling in macrophages, and these chemokines regulate the migration of innate immune cells during infection (*Griffith et al., 2014*; *Kuziel et al., 1997*). Similarly, expression of CD69, CD62L, and SWAP70, known to function in lymphocyte migration, are dependent on the

DDR signaling in developing B cells (*Bredemeyer et al., 2008*). Additionally, the DDR factor PAXIP1 was found to regulate egress of mature single positive T cells from the thymus (*Callen et al., 2012*). Taken together, these findings suggest that DNA DSBs and the resulting DDR influence the homing and migration of various immune cell types in a variety of distinct cellular contexts.

ATM and DNA-PKcs are both required for optimal IL-1$\beta$ production by macrophages in response to *L. monocytogenes* and diverse inflammasome activators. This is due, at least in part, to a role for the DDR in the conversion of pro-caspase 1 to active caspase-1. Interestingly, the production of IL-18, which is also generated through inflammasomes, depends on DNA-PKcs but not on ATM. Thus, ATM and DNA-PKcs may have distinct activities in the generation of IL-18 and IL-1$\beta$. The DDR in macrophages depends on type I IFN receptor signals. This is not due solely to the role of type I IFN in promoting the production of the DNA damaging agent, NO nor is it due to a requirement for type I IFN to promote DNA DSBs in response to DNA damaging agents. Rather, type I IFN is required to initiate the DDR to DSBs generated in macrophages. Although it is unclear how type I IFN enables at DDR in macrophages it does not regulate the expression of key DDR proteins (*Figure 4—figure supplement 2*). However, we find that many of these proteins are expressed at significantly lower levels than MEFs, which do not depend on type I IFN to initiate a robust DDR. Thus, it is conceivable that type I IFN activates pathways that permit robust DDR in cells expressing limiting amounts of DDR proteins. This could result from widespread chromatin modifications induced by type I IFN-dependent gene expression changes. Importantly, we find that this does not lead to an increase in DNA DSBs. However, these chromatin modifications may enhance DDR activation to DSBs (*Bakkenist and Kastan, 2003*; *Floyd et al., 2013*).

## Materials and methods

### Mice

All mice were bred and maintained under specific pathogen-free conditions at the Washington University School of Medicine and Weill Cornell Medical College under protocol number 2015–0036. Mice were handled in accordance with the guidelines set forth by the Division of Comparative Medicine of Washington University and the Research Animal Research Center at Weill Cornell Medical College. *Lyz2*$^{Cre/+}$ (*Clausen et al., 1999*), *Scid*, *Ifnar1*$^{-/-}$, *Tmem173*$^{-/-}$, and *Nox2*$^{-/-}$ mice were maintained on a C57BL/6 background. The *Atm*$^{-/-}$ and *Mre11*$^{ATLD1/ATLD1}$ mice have been described previously (*Theunissen et al., 2003*). Confirmation of mouse genotypes was done by PCR or Southern blotting. *Atm*$^{C/C}$ mice (*Zha et al., 2008*) were extensively backcrossed to the C57BL/6 background and were monitored by the analysis of microsatellite markers at the Rheumatic Disease Core Center, Washington University School of Medicine (St. Louis, MO). All mice were analyzed between 4 and 8 wks of age.

### Bacteria

The *Listeria monocytogenes* strain used in this study was the wild type strain EGD, which was stored as glycerol stocks at −80℃. For all ex vivo experiments, *L. monocytogenes* were grown and prepared as described previously (*Edelson and Unanue, 2001*). Heat-killed *L. monocytogenes* were prepared by incubation of mid-log bacteria at 70℃ for 3 hr followed by several washes with sterile 1X PBS.

### Cell culture

Bone marrow was harvested from 4–8-wk-old mice and cultured for 6 days in complete DMEM containing 10% heat-inactivated FBS, 5% heat-inactivated horse serum, and 20% culture supernatant from L929 fibroblasts as a source of M-CSF. BMDMs were re-plated in this media in 6-well plates at a density of 2.5 × 10$^6$/well. The following day, BMDMs were treated with 100 ng/mL LPS (*Escherichia coli* serotype 055:B5) (Sigma-Aldrich, St. Louis, MO), 100 U/mL murine IFN-γ (PBL Interferon Source), or both. For infection with *L. monocytogenes*, BMDMs were re-plated in antibiotic-free media and infected the following day with *L. monocytogenes* at a multiplicity of infection (MOI) of 5 followed by addition of Gentamicin (5 ug/mL) (Gibco) 30 min post-infection to kill extracellular *L. monocytogenes*. All *L. monocytogenes* experiments were done in the presence or absence of 100 ng/mL murine IFN-γ (R&D Systems). Primary peritoneal macrophages were harvested from C57BL/6

mice via peritoneal lavage. Cells were plated in 24-well plates at a density of 1–2 × 10⁶ cells/well and were incubated at 37°C for 4 hr in complete DMEM. At this time, media was removed and the remaining adherent macrophages were incubated for 24 hr in complete DMEM containing LPS (100 ng/mL), IFN-γ (100 U/mL), or both. Mouse embryonic fibroblasts (MEFs) were generated from C57BL/6 d 13.5 embryos and subsequently immortalized with the SV40 T antigen. BMDMs and mouse embryonic fibroblasts (MEFs) were treated with Bleocin (1 ug/mL) (Millipore) for 6 hr without or with IFN-$\beta$ (100 U/mL) (PBL Interferon Source) pretreatment for 3 hr. For inflammasome studies, BMDM were treated with 200 ng/mL LPS (*Escherichia coli* serotype 055:B5) (Sigma) with or without 5 ug/mL poly(dA:dT) (Invivogen) or monosodium urate crystals (MSU, 50 ug/mL) (Invivogen) for 12 hr or with LPS for 12 hr followed by nigericin (5 uM) (Invivogen) for 30 min. IL-1$\beta$ and IL-18 levels in supernatants were measured using the OptEIA ELISA set (BD Biosciences) and Mouse IL-18 ELISA Set (MBL), respectively, in accordance with the manufacturer's instructions. Supernatant nitrite levels were determined using the Griess reagent. Levels of mitochondrial superoxide were detected in live BMDM using the fluorogenic dye MitoSOX Red (2.5 uM) (Molecular Probes). Cells were treated with the dye for 30 min, washed with warm 1X PBS, and analyzed by flow cytometry using a BD LSR II (BD) and FlowJo software (TreeStar).

## PCR and southern blotting

RNA was isolated and quantitative RT-PCR carried out as described previously (*Bednarski et al., 2012*). Oligonucleotide sequences are detailed at the end of this section. Southern blot analyses were performed using KpnI-digested genomic DNA and the 3′ ATM conditional probe as previously described (*Bredemeyer et al., 2006*; *Zha et al., 2008*).

## Western blotting

BMDM and MEFs were lysed in RIPA buffer and whole cell lysates were generated with LDS sample buffer (Invitrogen) supplemented with dithiothreitol (DTT). For analysis of culture supernatants, protein was precipitated with 7.2% w/v trichloroacetic acid (TCA) (Sigma) followed by two acetone washes. Immunoblotting was carried out as previously described (*Helmink et al., 2011*). Primary antibodies used were anti-γ-H2AX clone JBW301 (Millipore) (RRID:AB_309864), anti-H2AX (Millipore) (RRID:AB_2233033), anti-phospho-KAP-1 (Bethyl Laboratories) (RRID:AB_669740), anti-KAP-1 (Gene-Tex) (RRID:AB_372041), anti-caspase 1 (p20, Casper-1) (Adipogen) (RRID:AB_2490248), anti-DNA-PKcs (Invitrogen), anti-Ku70 (Cell Signaling Technology), anti-Ku80 (Cell Signaling Technology) (RRID:AB_2257526), anti-ATM clone MAT3 (Sigma), anti-Mre11 (Novus) (RRID:AB_10077796), anti-Nbs1 (Abcam) (RRID:AB_777006), anti-Rad50 (Abcam) (RRID:AB_2176935), anti-ATR (Novus) (RRID: AB_10003234), and anti-glyceraldehyde-3-phosphate dehydrogenase (GAPDH) (Sigma). Secondary reagents were horseradish peroxidase–conjugated anti–mouse IgG (Promega) or horseradish peroxidase–conjugated anti- rabbit IgG (Cell Signaling Technology).

## Immunoflurescence

BMDM were plated on 12 mm glass coverslips (2.5 × 10⁵ cells/coverslip) in 24-well plates and infected with *L. monocytogenes* as described above. Nine hr post-infection, cells were fixed with 4% formaldehyde in PBS for 10 min at room temperature, permeabilized in 0.5% Triton X-100 in PBS for 5 min, and then washed with PBS. Coimmunostaining with primary and secondary antibodies was performed with a blocking solution of 3% bovine serum albumin (BSA) in PBS at 37°C for 30 min, and cells were mounted with ProLong Gold Antifade reagent containing 4′,6-diamidino-2-phenylindole (DAPI) (Invitrogen). Antibodies used for staining were anti-γ-H2AX clone JBW301(1:2000 dilution) (Millipore) (RRID:AB_309864), and Difco Listeria O Antiserum Poly Serotypes 1, 4 (1:200 dilution) (BD). Antibodies used for secondary staining were Alexa Fluor 488–goat anti-rabbit IgG (1:2000) (Invitrogen), and Alexa Fluor 594–goat anti mouse IgG (1:2000) (Invitrogen). Imaging was performed with a BX-53 Olympus microscope using an ApoN 60ˣ/1.49-numerical-aperture oil immersion lens and cellSens Dimension software.

## COMET assay

BMDM were treated with LPS and IFN-γ, infected with *L. monocytogenes*, or treated with Bleocin with or without IFN-$\beta$, as described above. Cells were subjected to the neutral CometAssay using

reagents from Trevigen in accordance with the manufacturer's protocol. Imaging was performed with a BX-53 Olympus microscope using an ApoN $60^X$/1.49-numerical-aperture oil immersion lens and cellSens Dimension software. Olive Tail Moment was determined using the software OpenComet. 50 or more tails were analyzed for each condition.

### Cell cycle analysis

BMDM were infected with *L. monocytogenes* for 16 hr as previously described or treated with Bleocin with or without IFN-$\beta$, as described above. Cells were then pulsed with BrdU for 30 min using the BrdU-FITC or –APC kits per the manufacturer's instructions. DNA content was assessed by 7AAD (BD) and data were acquired on a FACSCalibur or BD LSR II (BD Biosciences) and were analyzed with FlowJo software (TreeStar).

### Gene array

RNA was isolated from two independent BMDM cultures for each genotype (*Lyz2$^{Cre/+}$* and *Scid: Atm$^{C/C}$: Lyz2$^{Cre/+}$*) after no infection or infection with *L. monocytogenes* for 24 hr. RNA was extracted from cells with the RNeasy Mini Kit (Qiagen). Gene expression profiling was performed using Illumina MouseRef-8 expression microarrays according to the manufacturer's protocols. Unnormalized summary probe profiles were exported from GenomeStudio (Illumina) and background corrected and quantile normalized using the *limma* R package's (*Shi et al., 2010*) neqc function with default parameters. Only probes with a detection p-value<0.05 in at least three arrays were considered expressed and used for further analyses. A linear model was fit to the data, and an empirical Bayes moderated t-statistics test was used to determine differentially expressed genes in *L. monocytogenes*-infected cells relative to uninfected cells using the *limma* R package (*Ritchie et al., 2015*). Multiple probes for a given gene were resolved by retaining the probe with the highest average expression across all arrays. Upregulated genes with a fold change>4 and adjusted p-value<0.001 in *L. monocytogenes*-infected cells relative to uninfected were used for heatmaps. Fold changes were calculated based on the average of two biological replicates for each genotype.

### NK-BMDM co-culture assay

BMDM were cultured in 12-well plates at a density of $1 \times 10^6$ cells/well and infected with *L. monocytogenes* as described above. Splenic NK cells were magnetically sorted from whole splenocytes obtained from C57BL/6 mice using CD49b (DX5) MicroBeads and MS columns (Miltenyi). 12 hr post-*L. monocytogenes* infection of BMDM, $0.5 \times 10^6$ purified splenic NK cells were added to each well in the presence of IL-2 (50 U/mL) (PeproTech) to foster NK cell survival. Where relevant, murine IL-18 (10 ng/mL) (MBL) was also added to the cultures. 10 hr after adding the purified NK cells to the BMDM, protein transport was inhibited with GolgiStop (BD Biosciences). Five hr later, cells were harvested and nonspecific binding was blocked with 5 ug/mL of anti-CD16/32 (2.4G2; BD Pharmingen) before cell surfaces were stained with anti- NK1.1 (PK136; eBioscience), anti-CD11b (M1/70; BD Pharmingen), and anti-F4/80 (Invitrogen). Cells were fixed and permeabilized according to standard protocol and intracellular staining for IFN-$\gamma$ (XMG1.2, eBioscience) was performed. Data were acquired on a FACSCanto II (BD Biosciences) and were analyzed with FlowJo software (TreeStar).

### Statistical analysis

All p-values were generated via Student's unpaired two-tailed *t* test (unless otherwise stated in the figure legend) using Prism Version 5. P-values below 0.05 were considered statistically significant.

### Sequences of primers used in quantitative real-time PCR (RT-PCR) analysis

| | Forward primer | Reverse primer |
|---|---|---|
| Gapdh | AGGTCGGTGTGAACGGATTTG | TGTAGACCATGTAGTTGAGGTCA |
| Cxcl1 | CTGGGATTCACCTCAAGAACATC | CAGGGTCAAGGCAAGCCTC |
| Cxcl10 | CCAAGTGCTGCCGTCATTTTC | GGCTCGCAGGGATGATTTCAA |

| Ccl2 | TTAAAAACCTGGATCGGAACCAA | GCATTAGCTTCAGATTTACGGGT |
|------|-------------------------|-------------------------|
| Ccl3 | TTCTCTGTACCATGACACTCTGC | CGTGGAATCTTCCGGCTGTAG |
| Ccl4 | TTCCTGCTGTTTCTCCTCTTACACCT | CTGTCTGCCTCTTTTGGTCAG |
| Ccl5 | GCTGCTTTGCCTACCTCTCC | TCGAGTGACAAACACGACTGC |
| Marco | GCACAGAAGACAGAGCCGATTT | GCCACAGCACATCTCTAGCATCT |
| Cd69 | TGGTGAACTGGAACATTGGA | CAGTGGAAGTTTGCCTCACA |
| Il1b | AGCTTCCTTGTGCAAGTGTCT | GACAGCCCAGGTCAAAGGTT |
| Il18 | TCAAAGTGCCAGTGAACCCC | GGTCACAGCCAGTCCTCTTAC |
| Aim2 | CGGGAAATGCTGTTGTTGAC | TGCTCCTGGCAATCTGAAA |
| Il12p40 | ACCTGTGACACGCCTGAAGAAGAT | TCTTGTGGAGCAGCAGATGTGAGT |

## Accession numbers

The Gene Expression Omnibus number for the gene expression profiling analysis reported in this paper is GSE70467.

## Acknowledgements

We thank Dr. Emil R Unanue for stimulating our interest in examining DNA damage responses in macrophages and for thoughtful guidance throughout this project. We thank Drs. Vishva Dixit, Irma Stowe, Marina Cella, Victor Cortez, Daniel Graham, and Alejandro Reyes for helpful discussions and technical assistance. We thank Dr. Gwendalyn Randolph for critical review of the manuscript. We thank all of the members of our laboratory for many helpful discussions and Ryan D Irwin for maintaining the mouse colony. Experimental support was provided by the Speed Congenics Facility of the Rheumatic Diseases Core Center and the Genome Technology Access Center (Washington University School of Medicine, St. Louis, MO). This work was supported by the National Institutes of Health grants T32 AI007163 (AJM), R01 AI113118 (BTE) R01 AI047829 (BPS) and R01 AI074953 (BPS), a Burroughs Wellcome Fund Career Award for Medical Scientists (BTE), and by the Intramural Research Program of the NIH, National Institute of Environmental Health Sciences (Z01ES021157).

## Additional information

### Funding

| Funder | Grant reference number | Author |
|--------|------------------------|--------|
| National Institutes of Health | T32 AI007163 | Abigail J Morales |
| National Institutes of Health | R01 AI113118 | Brian T Edelson |
| National Institute of Allergy and Infectious Diseases | R01 AI047829 | Barry P Sleckman |
| National Institute of Allergy and Infectious Diseases | R01 AI074953 | Barry P Sleckman |
| National Institute of Environmental Health Sciences | Z01ES021157 | Barry P Sleckman |

The funders had no role in study design, data collection and interpretation, or the decision to submit the work for publication.

### Author contributions

AJM, PJH, Conceptualization, Resources, Data curation, Formal analysis, Supervision, Validation, Visualization, Methodology, Writing—original draft, Writing—review and editing; JAC, Conceptualization, Resources, Data curation, Formal analysis, Supervision, Funding acquisition, Investigation, Visualization, Methodology, Writing—original draft, Project administration, Writing—review and editing; ATT, Conceptualization, Resources, Data curation, Formal analysis, Supervision,

Visualization, Methodology, Writing—review and editing; JMA, Formal analysis, Validation; BTE, Conceptualization, Resources, Data curation, Formal analysis, Supervision, Validation, Investigation, Visualization, Methodology, Writing—review and editing; BC, Data curation, Formal analysis, Supervision, Methodology, Writing—review and editing; CLI, Conceptualization, Resources, Data curation, Formal analysis, Supervision, Investigation, Visualization, Methodology, Writing—review and editing; RSP, Conceptualization, Resources, Data curation, Formal analysis, Supervision, Validation, Methodology, Writing—review and editing; JEP, Resources, Data curation, Formal analysis, Supervision, Methodology, Writing—review and editing; BPS, Conceptualization, Resources, Formal analysis, Supervision, Funding acquisition, Investigation, Visualization, Methodology, Writing—original draft, Project administration, Writing—review and editing

### Author ORCIDs
Abigail J Morales, http://orcid.org/0000-0001-5148-4349
Barry P Sleckman, http://orcid.org/0000-0001-8295-4462

### Ethics
Animal experimentation: All mice were bred and maintained under specific pathogen-free conditions at the Washington University School of Medicine and Weill Cornell Medical College under protocol number 2015-0036. Mice were handled in accordance with the guidelines set forth by the Division of Comparative Medicine of Washington University and the Research Animal Research Center at Weill Cornell Medical College.

## Additional files

### Major datasets
The following dataset was generated:

| Author(s) | Year | Dataset title | Dataset URL | Database, license, and accessibility information |
|---|---|---|---|---|
| Sleckman BP, Morales AJ | 2015 | DNA damage responses activate a multi-functional genetic program in murine bone marrow-derived macrophages | https://www.ncbi.nlm.nih.gov/geo/query/acc.cgi?acc=GSE70467 | Publicly available at the NCBI Gene Expression Omnibus (accession no: GSE70467) |

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
