## [Decision Letter]

Thank you for submitting your article "A Type I IFN-Dependent DNA Damage Response Regulates the Genetic Program and Inflammasome Activation in Macrophages" for consideration by *eLife*. Your article has been reviewed by three peer reviewers, and the evaluation has been overseen by Michel Nussenzweig as the Reviewing and Senior Editor. The following individuals involved in review of your submission have agreed to reveal their identity: Rafael Casellas (Reviewer #3).

The reviewers have discussed the reviews with one another and the Reviewing Editor has drafted this decision to help you prepare a revised submission.

The referees were generally enthusiastic about the paper and made a number of suggestions that would help clarify the findings. These questions should be addressed in the revised manuscript when it is re-submitted.

1) Can the authors confirm that activated macrophages are in G0/G1?

2) Comment on how NO produces DSBs.

3) Comment on how they think type I IFN signaling promotes DDR initiation/DSBs independently of the NO production?

4) IFN-β increases gH2AX in bleo treated BMDMs but not Olive tail moment by COMET (4C,D). In contrast IFN-γ does not cause gH2AX increase in BMDM despite restoration of nitrite, indicating that type 1 interferon is necessary to mount an effective DDR. How does this work? Does IFN-b change the cell cycle or expression of DDR proteins necessary for gH2AX and KAP1 phosphorylation?

5) The authors should check expression of DDR genes ATM, ATR, DNA-PK, MRN, H2AX, +/- IFN-β. Does gene expression or protein levels of these factors account for the type 1 interferon requirement?

6) Chromatin structural changes are implicated in ATM activation (Bakkenist and Kastan, 2003; Floyd et al., 2013). Perhaps type 1 interferon is necessary to open up chromatin structure for PIKK activation and susbtrate phosphorylation. Does hypotonic or HDACi stimulate KAP-1 and gH2AX phosphorylation in Ifnar1-/- BMDM activated by l.m.?

7) The relationship between DNA breaks and transcriptional regulation is emerging in different systems. In this study, it remains unclear how these two cellular activities are linked. Are DNA brakes elicited at specific genomic locations?

8) Both IFN-β and IFN-γ can, along with LPS co-stimulation, drive macrophage activation and the DNA damage response, and yet loss of IFN-β signaling even in the presence of IFN-γ disrupts the process. Does the same hold true for IFN-γ?

9) The genes differentially expressed due to DNA damage receive only a brief discussion, and little analysis beyond quantification of the effect is performed. Is there an overlap between this gene group and the one seen in B cells downstream of DNA breaks?

---

## [Author Response]

*The referees were generally enthusiastic about the paper and made a number of suggestions that would help clarify the findings. These questions should be addressed in the revised manuscript when it is re-submitted.*

*1) Can the authors confirm that activated macrophages are in G0/G1?*

We now show in Figure 1—figure supplement 2 that wild type (Lyz2^Cre/+^) and DDR-deficient (Scid:Atm^C/C^:Lyz2^Cre/+^) macrophages arrest in G1 when infected with *ListeriaListeria monocytogenes*.

*2) Comment on how NO produces DSBs.*

NO has been shown to lead to the formation of single-strand nicks that, through unknown mechanisms, can form double-strand breaks. This is now discussed and referenced in the revised manuscript.

*3) Comment on how they think type I IFN signaling promotes DDR initiation/DSBs independently of the NO production?*

Our surprising finding that type I IFN signaling is required for a DDR in macrophages is something of great interest to us. The mechanistic basis of this requirement is unclear, but addressing the thoughtful comments of the reviewers has revealed important findings about this requirement that have been included in the revised manuscript. At the reviewers’ request, we investigated the potential effect of type I IFN on the cell cycle and the expression of key DDR proteins. We found that type I IFN has no effect on macrophage cell cycle (Figure 4—figure supplement 1) and does not alter the expression of the key DDR proteins H2AX, ATR, ATM, DNA-PK, Mre11, Rad50, Nbs1, Ku70 or Ku80 (Figure 4—figure supplement 2). We found, however, that compared to MEFs, macrophages express remarkably lower levels of most of these DDR proteins. Thus, it is conceivable that type I IFN functions to augment the DDR in settings where DDR protein expression is limiting. This possibility is now discussed in the revised manuscript.

*4) IFN-β increases gH2AX in bleo treated BMDMs but not Olive tail moment by COMET (4C,D). In contrast IFN-γ does not cause gH2AX increase in BMDM despite restoration of nitrite, indicating that type 1 interferon is necessary to mount an effective DDR. How does this work? Does IFN-b change the cell cycle or expression of DDR proteins necessary for gH2AX and KAP1 phosphorylation?*

See answer to comment 3 above.

*5) The authors should check expression of DDR genes ATM, ATR, DNA-PK, MRN, H2AX, +/- IFN-β. Does gene expression or protein levels of these factors account for the type 1 interferon requirement?*

See answer to comment 3 above.

*6) Chromatin structural changes are implicated in ATM activation (Bakkenist and Kastan, 2003; Floyd et al., 2013). Perhaps type 1 interferon is necessary to open up chromatin structure for PIKK activation and susbtrate phosphorylation. Does hypotonic or HDACi stimulate KAP-1 and gH2AX phosphorylation in Ifnar1-/- BMDM activated by l.m.?*

This is an excellent point. Initially we thought that perhaps the induction of gene expression by type I IFN may lead to a global increase in accessibility that would increase the generation of DNA DSBs, but the COMET assay revealed that this is not the case. However, as the reviewer points out, it may be that these chromatin changes permit more efficient activation of the DDR in macrophages treated with type I IFN. This may be especially important given that the levels of DDR proteins appear to be significantly lower in macrophages as compared to MEFs (see response to point 3 above). Interrogating this possibility experimentally will certainly be a focus of future studies in this area. Nevertheless, we now discuss this important possibility in the revised manuscript and thank the reviewer for suggesting this.

*7) The relationship between DNA breaks and transcriptional regulation is emerging in different systems. In this study, it remains unclear how these two cellular activities are linked. Are DNA brakes elicited at specific genomic locations?*

This is an excellent question. These DSBs are not programmed DNA DSBs where the genomic locations of the DSB can be determined, like those made by RAG during V(D)J recombination or AID during Ig class switch recombination. Rather, the DSBs made by NO are likely random throughout the genome like those made by ionizing radiation and radiomimetic drugs. This is discussed more clearly in the revised manuscript.

*8) Both IFN-β and IFN-γ can, along with LPS co-stimulation, drive macrophage activation and the DNA damage response, and yet loss of IFN-β signaling even in the presence of IFN-γ disrupts the process. Does the same hold true for IFN-γ?*

We have shown (Figure 4) that type 1 IFN, but not IFN-γ, augments the DDR in macrophages treated with bleomycin.

9) The genes differentially expressed due to DNA damage receive only a brief discussion, and little analysis beyond quantification of the effect is performed. Is there an overlap between this gene group and the one seen in B cells downstream of DNA breaks?

We have expanded the discussion of the gene expression changes and the potential overlap with our previously published data in B cells.